# A novel multi-user collaborative cognitive radio spectrum sensing model: Based on a CNN-LSTM model

**Kai Wang, Yangyang Chen, Dan Bo, Shubin Wang**[ID] *

School of Electronic Information Engineering, Inner Mongolia University, Hohhot, Inner Mongolia, China

* MagicStroke8m2@mail.com

**Data Availability Statement:** All relevant data are within the manuscript and its Supporting Information files.

**Funding:** The author(s) received no specific funding for this work.

## Abstract

Cognitive Radio (CR) technology enables wireless devices to learn about their surrounding spectrum environment through sensing capabilities, thereby facilitating efficient spectrum utilization without interfering with the normal operation of licensed users. This study aims to enhance spectrum sensing in multi-user cooperative cognitive radio systems by leveraging a hybrid model that combines Convolutional Neural Networks (CNN) and Long Short-Term Memory (LSTM) networks. A novel multi-user cooperative spectrum sensing model is developed, utilizing CNN's local feature extraction capability and LSTM's advantage in handling sequential data to optimize sensing accuracy and efficiency. Furthermore, a multi-head self-attention mechanism is incorporated to improve information flow, enhancing the model's adaptability and robustness in dynamic and complex environments. Simulation experiments were conducted to quantitatively evaluate the performance of the proposed model. The results demonstrate that the CNN-LSTM model achieves low sensing error rates across various numbers of secondary users (16, 24, 32, 40, 48), with a particularly low sensing error of 9.9658% under the 32-user configuration. Additionally, when comparing the sensing errors of different deep learning models, the proposed model consistently outperformed others, showing a 12% lower sensing error under low-power conditions (100 mW). This study successfully develops a CNN-LSTM-based cooperative spectrum sensing model for multi-user cognitive radio systems, significantly improving sensing accuracy and efficiency. By integrating CNN and LSTM technologies, the model not only enhances sensing performance but also improves the handling of long-term dependencies in time-series data, offering a novel technical approach and theoretical support for cognitive radio research. Moreover, the introduction of the multi-head self-attention mechanism further optimizes the model's adaptability to complex environments, demonstrating significant potential for practical applications.

## 1. Introduction

In recent years, with the rapid development of wireless communication technologies, the efficient utilization of spectrum resources has become a research hotspot. Early studies revealed

**Competing interests:** The authors have declared that no competing interests exist.

that during periods or in regions with low user density, many licensed spectrum bands remain underutilized, resulting in significant spectrum resource wastage [1]. To address this issue, Joseph Mitola first proposed the concept of Cognitive Radio (CR) [2], which was further extended into the development of cognitive radio networks [3]. Cognitive radio technology enables wireless devices to learn about their surrounding spectrum environment through sensing capabilities, identify available spectrum resources, and allow secondary users to access these bands without interfering with the normal operation of primary users (PUs), thereby facilitating spectrum sharing [4].CR technology effectively addresses the aforementioned spectrum inefficiency, with spectrum sensing serving as the foundation and prerequisite for its applications. The primary task of spectrum sensing is to understand the spectrum usage of PUs in a specific communication area, making it the most critical step in the cognitive cycle [5]. The success of cognitive radio depends on the accuracy and efficiency of spectrum sensing. Enhancing the reliability and efficiency of sensing, minimizing interference with PUs, and enabling secondary users to quickly and accurately identify vacant spectrum to gain more access opportunities are key to improving the overall performance of cognitive radio networks [6]. This is crucial for reducing spectrum resource wastage and achieving the efficient utilization of spectrum, making it a focal point in the field of cognitive radio research.

Spectrum sensing, as the core enabling technology of cognitive radio (CR), primarily functions to dynamically evaluate the radio environment and identify underutilized spectrum opportunities [7]. Current research efforts have largely focused on improving the efficiency of spectrum sensing, with the majority targeting single-user spectrum sensing strategies, including energy detection [8], matched filter detection [9], and cyclostationary feature detection [10]. While these single-user approaches can quickly determine whether the primary user (PU) is active, the performance of spectrum sensing by a single secondary user (SU) is often vulnerable to adverse environmental changes [11]. To address these limitations, researchers have proposed cooperative spectrum sensing (CSS) strategies, wherein multiple SUs within the same cognitive radio network (CRN) collaborate to perform spectrum detection tasks [12]. CSS significantly enhances the efficiency of spectrum resource utilization in wireless networks [13]. However, this approach also introduces challenges in balancing performance improvements for both PUs and SUs. For instance, when SUs move continuously and randomly at uniform speeds, variations in their geographic locations, inter-SU distances, and communication environments affect spatial correlation among SUs, thereby impacting the stability and reliability of cooperative spectrum sensing [14]. Additionally, while traditional convolutional neural networks (CNNs) have shown promise in processing correlated data, their learning capabilities remain limited. These models still require further improvements to fully enhance spectrum sensing performance.

In light of the aforementioned challenges, this study proposes the PCBM (Parallel CNN_BiLSTM_MHSA) collaborative cognitive radio spectrum sensing model based on a CNN-LSTM architecture to enhance the spectrum sensing capability of multi-user collaborative cognitive radio systems. The model integrates Convolutional Neural Networks (CNNs), Bidirectional Long Short-Term Memory networks (BiLSTMs), and Multi-Head Self-Attention mechanisms (MHSA).

By leveraging the CNN's strength in local feature extraction and the superior performance of LSTMs in processing sequential data, the proposed model synergizes local and global feature extraction networks to capture multi-scale spatial characteristics and temporal sequence features of spectrum signals. MHSA further enhances the feature representation capability. Through a multi-user collaboration mechanism, the model optimizes spectrum sensing performance, demonstrating significant improvements in accuracy and robustness under complex signal environments. Compared to previous studies, this research is the first to propose a

cognitive radio spectrum sensing model that incorporates multi-user collaboration, overcoming the limitations of traditional single models in spectrum sensing and enhancing the effectiveness of spectrum utilization. The introduction of the MHSA mechanism further optimizes the flow of information within the network, improving the model's stability and adaptability in dynamic and complex environments. The main contributions of this study include: 1) Developing an efficient deep learning model that provides technical support for deploying cognitive radio systems; 2) Optimizing system performance through intelligent collaboration strategies, offering a novel approach to rational spectrum resource allocation; 3) Providing theoretical support and a technical pathway for the formulation and updating of spectrum management policies.

## 2. Progress in related research

### 2.1. Spectrum sensing technology

Spectrum sensing, as the core of cognitive radio (CR) technology, can help CR systems promptly identify available idle frequency bands and ensure that such utilization does not affect the PU's use of the spectrum. The primary goal of spectrum sensing is to identify and utilize the so-called "spectrum holes" to achieve efficient spectrum utilization [15].

Specifically, the task of spectrum sensing can be summarized into two main aspects: first, detecting whether a PU signal exists in a given frequency band and evaluating the status of the band to determine whether secondary users (SUs), i.e., unlicensed users, can use the band. When SUs require communication, the detected idle frequency bands can be allocated for their use [16]. Second, CR systems aim to improve the utilization of spectrum resources under the condition of not causing significant interference to PU communications, which requires SUs to quickly and accurately sense the presence of PUs. When licensed bands are idle, SUs should detect the idle channels as quickly as possible and continuously monitor during use for the reappearance of PUs so that they can vacate the bands promptly for PU use [17].

Over the past 20 years, numerous sensing algorithms for detecting spectrum holes have emerged, and these algorithms can be classified from multiple perspectives. Current spectrum sensing technologies are mainly divided into narrowband spectrum sensing and wideband spectrum sensing. Narrowband spectrum sensing typically filters out specific frequency bands using a band-pass filter after receiving the signals, defining the spectrum sensing problem as a binary hypothesis problem [18]. Traditional narrowband spectrum sensing methods include energy detection (ED), matched filter detection (MFD), cyclostationary feature detection (CFD), and autocorrelation detection, among others [19]. Wideband spectrum sensing, on the other hand, extends the detection range based on narrowband sensing, not only determining the presence of PU signals but also estimating the frequency bands occupied by PUs [20].

### 2.2. Multi-user spectrum sensing methods

Collaborative spectrum sensing (CSS) is a process where multiple SUs exchange information and share data within a network environment to mitigate potential adverse effects that individual SUs might encounter during the sensing process [21]. Currently, based on the method of information exchange among SUs and the criteria for information fusion, CSS systems can be broadly divided into two categories.

Centralized collaborative sensing systems consist of primary users, multiple secondary users, and a data fusion center. The central module in this system acts as the fusion center (FC), responsible for controlling the entire collaborative sensing process [22]. In this mode, the FC first selects specific frequency bands for sensing, assigns individual SUs to perform sensing tasks independently, and collects the decision statistics or sensing data from various

SUs. The FC then makes its binary decision, which is sent back to the FC. Based on a predefined fusion rule, the FC consolidates this information to determine the presence of a PU and communicates the decision back to all SUs. This approach is widely adopted due to its clear system design, fast operation, and strong real-time responsiveness. However, its main drawback is the reliance on the fusion center for data integration. A failure at the FC can disrupt the ability of all SUs to use the frequency band [23].

In a distributed collaborative sensing mode, all SUs independently perform sensing and then share their results with other CR users. Each SU combines its sensing results with those of other SUs, according to predefined fusion rules, to decide whether a PU is present [24]. This mode's advantage lies in its independence from the FC for decision-making, as information sharing and processing occur solely among SUs, saving signal transmission time. Additionally, distributed sensing reduces reliance on central infrastructure, leading to lower costs. However, this approach requires each SU to share and analyze sensing information in real-time to make decisions, which increases system complexity, reduces efficiency, and imposes higher hardware requirements [25]. Distributed collaborative algorithms are still under development and may not match the sensing performance of centralized collaborative sensing.

In CSS systems, based on the type of transmitted data, FC decision fusion can be divided into soft decision and hard decision approaches [26]. Hard decision fusion involves each SU using sensing techniques to detect the PU spectrum status and determining the presence of PU signals. Under hard decision rules, each SU performs a local decision and sends only a binary result (1 for PU present, 0 for PU absent) to the FC. The FC then fuses these local decisions based on predefined criteria and broadcasts the final decision to all SUs [27]. In soft decision fusion, each SU collects information about the licensed spectrum using various spectrum sensing algorithms. The collected information is transmitted to the FC, where the FC analyzes the data from all SUs and independently evaluates the PU signal's presence. Based on this evaluation, the FC forms a final decision and broadcasts the result to all SUs [28].

In recent years, significant progress has been made in the field of collaborative spectrum sensing through the introduction of machine learning techniques. For example. Shi et al. (2020) utilized the energy vectors of received signals as features to develop several machine learning-based spectrum sensing algorithms, including support vector machines (SVM), weighted K-nearest neighbors (WKNN), K-means clustering, and Gaussian mixture models. Experimental validation demonstrated that machine learning-based algorithms adapt more effectively to environmental changes and exhibit superior sensing performance [29]. Lu et al. (2016) proposed a novel collaborative spectrum sensing algorithm that combines K-means clustering and SVM by using low-dimensional probability vectors as inputs [30]. Additionally, Ghazizadeh and his team considered that the front-end sensing devices of unlicensed users often employ multi-antenna technology. By extracting eigenvalues from the sample covariance matrix, they applied an improved SVM algorithm for collaborative spectrum sensing, achieving better results than traditional linear kernel SVM algorithms [31]. Although machine learning techniques have shown promising applications in collaborative spectrum sensing, research on deep learning in this field remains in its early stages. Lee et al. (2019) developed a CNN-based collaborative spectrum sensing algorithm specifically designed for scenarios with multiple channels and randomly moving SUs. By exploring spatial correlations among SUs, the algorithm performed collaborative spectrum sensing [32]. Nesraoui et al. (2024) proposed a robust method called DET-AMC (Detection and Automatic Modulation Classification), which utilizes a convolutional neural network (CNN) trained via transfer learning. The CNN features obtained through transfer learning demonstrate robustness, particularly under low SNR conditions and various challenging scenarios, enabling accurate modulation classification [33]. Khichar et al. (2024) introduced a Fast Super-Resolution Convolutional Neural Network

(FSRCNN) model for channel estimation, aiming to reduce computational complexity while maintaining high estimation accuracy [34].Under varying noise levels and different numbers of SUs, the sensing performance of this algorithm surpassed that of traditional and machine learning-based collaborative spectrum sensing algorithms, providing strong evidence for the application of deep learning techniques in this domain.

## 3. Collaborative cognitive radio spectrum awareness modeling

### 3.1. System modeling

Addressing the problem of centralized collaborative spectrum sensing, this paper explores cognitive radio (CR) systems covering multiple sub-users (SUs) and multiple channels. It is envisioned that a single primary user (PU) and $N_{SU}$ SUs are randomly distributed in a square area of 200-meter side length, and all users move at a certain speed $v$ with a uniform speed in random directions. The location of the fusion center (FC) within the region is fixed. Due to the mobility of the users, the geographic locations of the SUs and their relative distances are constantly changing, so the indexing of the SUs is assigned based on the temporal order of their participation in the collaborative spectrum sensing. It is assumed that the authorized spectral bandwidth of the system is $W$, the number of channels is $N_{\tan d}$, the transmission power of the PU is set to P, and the energy leakage factor is set to 7. Under normal circumstances, the PU communicates mainly on one channel, but the transmission energy may leak to adjacent channels, resulting in the PU potentially affecting two or more neighboring channels at the same time. In practical radio communication scenarios, affected by noise and other factors, the PU may switch to other channels using frequency hopping techniques. In order to prevent interference with the PU's communication, once the PU signal is detected, all channels within the authorized spectrum should be considered unavailable for the SU. Therefore, this collaborative spectrum perception problem can be constructed as a binary hypothesis testing problem, which can further be modeled by deep learning methods for binary classification.

The additive Gaussian white noise with the power spectral density of $N_0$ is also taken into account in this study, and $\omega_i^j(m)$ is set to represent the noise of the ith SU in the jth channel at the moment m. The exponent and constant of the path loss are denoted as α and β. Set $d_i m$ to represent the distance between the PU and the ith SU at moment m. The path loss can be expressed as $\beta(d_i(m))^{\alpha}$. Set $g_i^j(m)$ to represent the multipath fading of the ith SU in the jth channel at moment m. In this paper, $g_i^j(m)$ is modeled as an independent circularly symmetric complex Gaussian random variable with zero mean.

Meanwhile, considering the case of shadow fading, set $h_i m$ to denote the shadow fading of PU between moment m and the ith SU, obeying the normal distribution with zero mean and zero variance $\sigma^2$ in dB. Set $k_i m$ to denote the normalized shadow fading of the ith SU at moment m, possessing the zero mean and unit variance, which can be computed by Eq (1).

$$k_i(m) = \sqrt{\frac{P}{\beta(d_i(m))^{\alpha} 10^{\frac{h_i(m)}{10}}}} \tag{1}$$

Assuming that the distance between SUA and SUB is d-, the correlation of normalized shadow fading between the two can be expressed as $\rho_{\text{cor}}(d_{A-B})$, and the computational procedure can be represented:

$$\rho_{\text{cor}}(d_{A-B}) = \text{E}[k_A(m)k_B(m)] = e^{\left(-\frac{d_{A-B}}{d_{ref}}\right)} \tag{2}$$

Where $d_{A-B}$ denotes a reference distance that depends on the environment in which it is located. Two SUs at a similar distance experience more similar shadow decay.

During each sensing cycle $T_p$, each SU performs energy detection on all $N_{band}$ channels and collects $N_{ED}$ samples for local sensing. This approach allows each SU to independently evaluate the state of the channels, enhancing the accuracy of channel state detection. To ensure the efficient utilization of spectrum resources without interfering with PUs, this study adopts a two-layer energy detection mechanism to distinguish between PU and SU signals. In the first layer, the system sets an energy detection threshold $E_{th}$ to identify PU signals. When the detected channel energy $E \geq E_{th}$, the system determines that the channel is occupied by a PU, thereby preventing SUs from communicating on that frequency band. If the channel energy is below the threshold, the system further analyzes the frequency characteristics of the channel signal. By extracting spectrum information, it verifies whether the signal conforms to the transmission pattern of a PU. Considering the synchronization requirements of SUs, to ensure that all SUs complete spectrum sensing operations within the same time window, SUs are allowed to coordinate their overall sensing time using a clock synchronization protocol during the spectrum sensing cycle. This enables the sharing of detection information, reducing the risk of misjudgments caused by data inconsistencies among different SUs. It also ensures that spectrum detection and the transmission of detection results are completed within their respective time slots, achieving efficient collaborative sensing.

In this study, the utilization state of the licensed spectrum is divided into two hypotheses. Hypothesis $H_0$ indicates that the licensed spectrum is in an idle state, meaning the PU is not utilizing the spectrum. Hypothesis $H_1$, on the other hand, indicates that the licensed spectrum is being utilized by the PU. These two hypotheses form the foundation of spectrum sensing and determine how the SU acts based on the sensed information. The received signal data $y_i^j(m)$ of the $i$-th SU at time $m$ on the $j$-th channel is expressed as:

$$y_i^j(m) = \begin{cases} k_i(m)g_i^j(m)x(m) + \omega_i^j(m), H_1, \ j \in B_P \\ \sqrt{\eta}k_i(m)g_i^j(m)x(m) + \omega_i^j(m), H_1, \ j \in B_A \\ \omega_i^j(m), H_0 \end{cases} \tag{3}$$

In the formula, $x(m)$ represents the signal data transmitted by the PU at time $m$, $B_P$ denotes the set of channels utilized by the PU, and $B_A$ represents the set of channels affected by leaked energy from PU communications.

The local sensing of all SUs is based on energy detection, where the signal energy intensity in the channel is obtained through cumulative calculation. On this basis, let $T_i^j$ represent the cumulative signal energy of the $i$-th SU on the $j$-th channel. The calculation formula is as follows:

$$T_i^j = \frac{1}{N_{ED}} \sum_{m=1}^{N_{ED}} |y_i^j(m)|^2 \tag{4}$$

## 3.2. Model structure

### 3.2.1. Localized feature extraction network structure.
With the significant enhancement of computational resources, convolutional neural networks have demonstrated exceptional capabilities especially in areas such as image processing and computer vision, and are particularly good at extracting local features. When dealing with image data, two-dimensional convolutional neural networks (2DCNN) are usually employed to deal with this type of data in

matrix form. For the classification task of sequence data, a one-dimensional convolutional neural network (1DCNN) is preferred, which is effective in extracting dense local features from fixed-length segments of sequence data and analyzing the spectral relationships between individual sequence channels. The 1D convolution shows advantages over the 2D convolution in terms of smaller number of parameters and faster training speed. The network design consists of three convolutional modules and an Average Pooling (AvePooling) layer. Each convolutional module is the underlying local feature extraction unit, which contains a convolutional layer (Conv1D), a Batch Normalization (BN) layer, a LeakyReLU activation function, and a Dropout layer, which work together to capture local features from the input data. In each Conv1D layer, there are multiple convolutional feature signals, and each convolutional feature signal C(m = 1,2. . ., M) are connected to the multivariate sequential input signal 7th univariate sequential data S,(1 = 1,2. . .) by a fixed weight matrix W of the form LxF., L). Where L denotes the number of variables also known as the number of SUs Nu in this chapter and F denotes the size of the one-dimensional convolution kernel. The size of the convolution kernel determines the number of elements of the input signal to which a single element of each convolutional feature signal is connected, i.e., the size of the sensory field. The corresponding mapping operation is the so-called convolution operation. Assume that each convolutional feature signal has k elements per convolutional feature signal C. Each element is calculated by Eq (5).

$$c_{m,k} = \sigma\left(\sum_{l=1}^{L}\sum_{f=1}^{F} s_{l,f+k-1} w_{l,m,f}\right) \tag{5}$$

In the above equation, σ is the nonlinear activation function; $c_{m,k}$ is the kth element of the mth convolutional feature signal $C_m$; $S_{l,k}$ is the kth element of the lth univariate sequence data of the multivariate sequence input signal $S_l$; $w_{l,m,f}$ is the fth element of the weight matrix $W_{l,m}$, and the weight matrix $W_{l,m}$ connects the lth univariate sequence data of the multivariate sequence input signal, mapping to the mth convolutional feature signal of the convolutional feature signal, which can be simplified on the basis of this further:

$$C_m = \sigma\left(\sum_{l=1}^{L} S_l * W_{l,m}\right) \tag{6}$$

In the field of deep learning, commonly used nonlinear activation functions include the hyperbolic tangent function (Tanh), the S-shaped function (Sigmoid), and the linear rectification unit (ReLU). ReLU, in particular, helps to enhance the sparsity of the model and reduce the risk of overfitting to some extent because it sets the output of neurons with negative outputs to zero. However, this activation function also has its limitations, such as causing the disappearance of the negative gradient, which may lead to the permanent inactivation of some neurons, i.e., the so-called "neuron death" phenomenon. In addition, ReLU may be overly enforced in the implementation of sparsification, which may sometimes lead to the loss of key feature information, thus reducing the effective capacity of the model. To solve this problem, we adopt the LeakyReLU activation function as the activation function σ of the convolutional layer, which is designed to allow a small gradient to pass through even when the output is negative, ensuring that the neuron maintains the gradient transfer in all situations. Its expression is given as:

$$f(x) = \begin{cases} \alpha x, x < 0 \\ x, x \geq 0 \end{cases} \tag{7}$$

In the design of LeakyReLU activation function, for the case that the input signal x is less than 0, the function does not set the output to 0 completely, but adopts a small coefficient α to correct the negative value, so as to retain the negative gradient, which can help to alleviate the loss of feature information caused by ReLU. In this way, LeakyReLU can improve the "necrosis" phenomenon of nerve cells and the gradient disappearance problem, and thus speed up the training speed of the model and improve the perceptual performance of the model.

In addition, in this paper, a Batch Normalization (BN) layer and a Dropout layer are added after each one-dimensional convolutional (Conv1D) layer. The BN layer helps to eliminate the internal covariate bias by adjusting the distribution of the input layers during the training process, making the neural network training more stable, and effectively mitigating the problem of gradient explosion, thus accelerating the The Dropout layer avoids overdependence of the model on specific neurons by randomly dropping a portion of the neurons in the network, which enhances the robustness of the model, helps to prevent overfitting, and improves the generalization ability of the model.

An average pooling layer is employed after the three convolutional modules of the model, which reduces the dimensionality of the feature signals and increases the invariance of the features through an average pooling operation in order to reduce the impact of fluctuations in the input data on the model. Specifically, the kth element of the mth pooled feature signal $P_m(m = 1,2,\ldots,M)$ of the average pooling layer can be expressed as:

$$p_{m,k} = r \sum_{n=1}^{N}(c_{m,(k-1)\times q+n}) \tag{8}$$

where $N$ is the pooling size, $q$ is the sliding step size, and $r$ is the scaling factor.

**3.2.2. Global feature extraction network structure.** When dealing with multivariate serial data classification problems, the interactions and correlations among different variables need to be considered. The mechanism of multiple self-attention (MHSA) is particularly suitable for detecting the complex interrelationships among variables in such data. MHSA is able to sensitively capture the direct or indirect correlations among variables in multivariate sequence data and reveal the correlations among features by correlating different positions in the sequences, which is useful for identifying the interactions among hidden features. In addition, MHSA helps to compensate for the loss of information that may be overlooked during the feature extraction process. Compared with the traditional single-head self-attention mechanism, MHSA is able to improve the ability to perceive complex correlations among multivariables and mine richer feature relationships more effectively by processing multiple attention heads in parallel. The self-attention mechanism (SA) is implemented by calculating the similarity between the input Query and Key to assign different weight coefficients, and then weighting and summing the Value according to these weights. This mechanism allows the model to dynamically extract the most relevant information for the task at hand from the entire sequence when processing sequential data, where Queries, Keys and Values are usually derived from the same input data. This mechanism is designed to make MHSA particularly suitable for processing multivariate sequence data that have complex interactions between variables, thereby improving the overall performance and prediction accuracy of the model.

As shown in Fig 1, from the viewpoint of network structure, MHSA is splicing the outputs of multiple SAs and then doing parametric transformations, so that $Head_i$ denotes the output of the SA with the i-th head, then the output of MHSA with the number of heads h can be expressed:

$$MultiHead = Concat(Head_1, Head_2, \ldots, Head_h)W_o \tag{9}$$

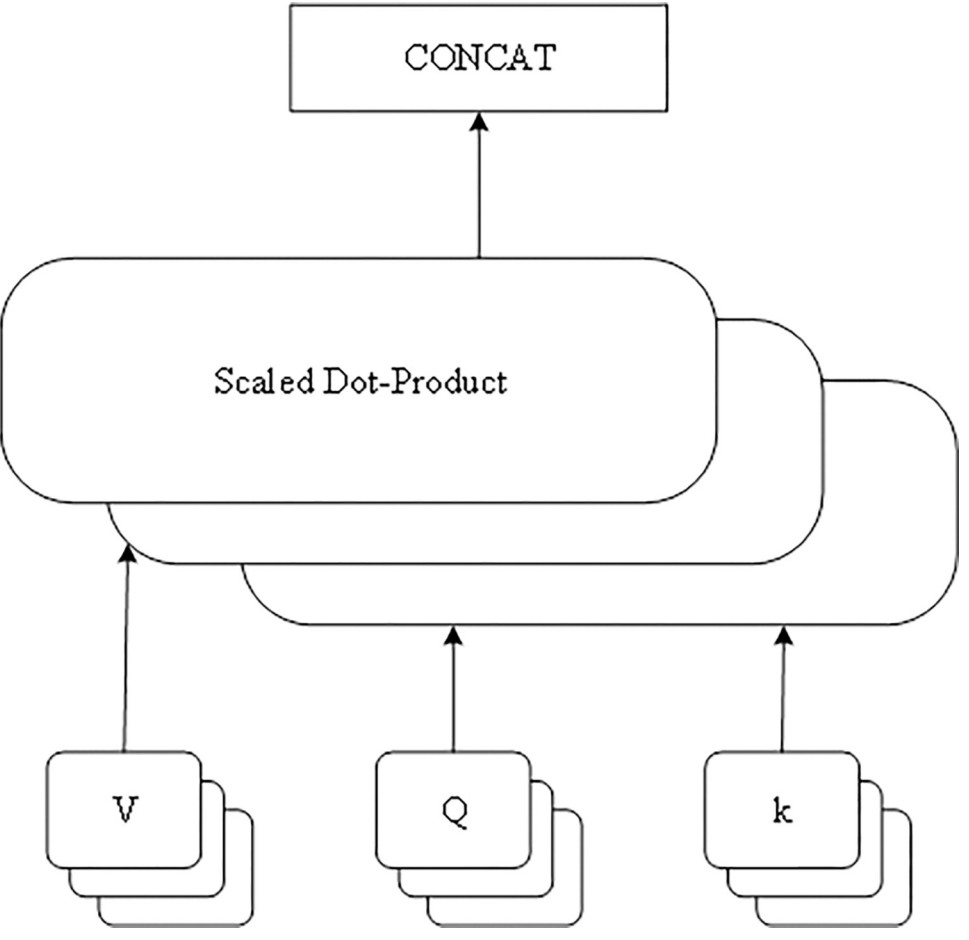

**Fig 1. Structure of the multi-leader self-attention mechanism.**

where MuiHead is the output of MHSA, Concat is the Concatenated splicing layer in deep learning, and $W_o$ denotes the weight matrix for parameter transformation of the splicing results output from each SA.

**3.2.3. Training methods.** In the dual-network architecture, the local feature extraction network (1D-CNN) and the global feature extraction network (BiLSTM) perform their respective functions before merging their output signals through a concatenation layer. The local feature extraction network employs multiple convolutional layers to capture the local multi-scale features of spectrum signals, while the global feature extraction network further captures the global temporal dependencies of the sequences from the local features. The merged signal is then passed to the classification network to produce the final classification decision. At the end of the classification network, a fully connected layer with two neurons and a Softmax activation function outputs the classification probabilities. During training, the model updates its parameters using a joint optimization strategy. The training data first pass through the local feature extraction network, generating multi-dimensional feature maps. These feature maps are then forwarded to the global feature extraction network to learn the global characteristics of the time series. Subsequently, the local and global features are concatenated into a unified feature vector, which is passed to the classification network. The final prediction error is computed using a cross-entropy loss function, and the parameters of both the local and global networks are simultaneously updated via backpropagation using the Adam optimizer.

The loss function of the model is chosen as cross-entropy, which evaluates the error $L_{super}$ between the predicted results and the actual labels. The calculation formula is as follows:

$$L_{super} = -\frac{1}{U}\sum_{u=1}^{U}\left(\hat{Y}_u\log(p_u)\right) \tag{10}$$

In the formula, $\hat{Y}_u$ and $p_u(u = 1,2,\ldots,U$ represent the true label and the corresponding predicted result of the u-th training sample, respectively, and U denotes the total number of training samples.

This strategy leverages the advantages of a dual-network model to extract deeper features with fewer parameters, thereby optimizing the classification performance of a single-network algorithm. At the same time, to ensure training efficiency and superior model performance, the model adopts the following hyperparameter strategies:(1) Batch Size: Considering the memory capacity limitations of the hardware, the batch size is set to 2000, balancing memory constraints and improving training stability.(2) Dynamic Learning Rate Adjustment: Additionally, the model employs an appropriate initial learning rate combined with a dynamic decay strategy to avoid issues where a learning rate that is too high prevents convergence, or a rate that is too low significantly slows down training. The initial learning rate is set to 0.003, combined with an exponential decay strategy, to mitigate the convergence issues caused by an excessively high learning rate and the efficiency decline due to an overly low learning rate.(3) Overfitting Prevention: To further prevent overfitting, the model incorporates L2 regularization, Dropout with a dropout rate of 0.3, and an early stopping strategy with a patience parameter of 5, thereby optimizing the model's generalization capability. Additionally, for the update of model weights and biases, the Adaptive Moment Estimation (Adam) optimizer is employed, with Adam's parameters set as follows: $\beta_1 = 0.9$, $\beta_2 = 0.999$, $\varepsilon = 10^{-8}$, to ensure the training process is efficient and the final model performance is superior. This comprehensive parameter tuning and algorithm application provide a solid foundation for the optimization and performance improvement of the model.

# 4. Findings and analysis

## 4.1. Parameterization

To evaluate the performance of the proposed model, this study employs simulation experiments to optimize neural network parameters, ensuring optimal sensing performance. In the cognitive radio system used for the experiments, user density is reflected by variations in the number of secondary users (SUs). The number of SUs is set at five levels: 16, 24, 32, 40, and 48. Environmental noise levels are indirectly simulated through the transmitter power range (100–1000 mW). Based on the relationship between transmitter power and noise levels, it is observed that lower transmission power results in a lower signal-to-noise ratio (SNR), representing more complex noise environments. Conversely, higher transmission power increases the SNR, approximating ideal channel conditions [35]. This approach simplifies the complexity of directly simulating noise while providing an effective method for assessing the model's adaptability to complex spectrum environments. The primary user's (PU) transmission power is increased from 100 mW to 1000 mW in 100 mW increments. For each power condition, 200,000 data samples are generated, totaling 2,000,000 samples. These data are divided into training, validation, and test sets in a 7:1:2 ratio. In the design of the local feature extraction network, the 1D convolutional neural network (1DCNN) consists of three Conv1D layers, each with 128 convolutional kernels. The three-layer convolutional structure enables the network to progressively extract multi-level features of the spectrum signals, capturing high-

frequency features while extracting low-frequency features at deeper levels, thereby improving sensing accuracy [36]. Meanwhile, the use of 128 convolutional kernels strikes a good balance between high- and low-frequency feature extraction for the spectral data. Studies have shown that an appropriate number of convolutional kernels can enhance feature extraction capability without significantly increasing computational complexity [37]. In this study, Sensing Error is adopted as the core metric to evaluate model performance. Sensing error measures the discrepancy between the predicted and actual categories in the spectrum sensing task, defined as follows:

$$\text{Sensing Error} = \frac{1}{N}\sum_{i=1}^{N}|\hat{y}_i - y_i| \tag{11}$$

Where N represents the total number of test samples, $\hat{y}_i$ denotes the predicted category of the i-th sample, and $y_i$ is the corresponding ground truth category. A smaller sensing error indicates higher classification accuracy, making it an effective metric for evaluating model performance. The choice of sensing error as the evaluation metric is based on its intuitiveness, adaptability, and comparability. As a direct measure of classification accuracy, sensing error clearly reflects the model's performance under varying experimental conditions. Moreover, in cognitive radio networks, environmental factors such as noise levels and user density significantly affect classification accuracy. Sensing error robustly captures the model's performance across different environmental conditions. Defined based on absolute error, sensing error simplifies comparisons between different models, facilitating the validation of the proposed model's advantages in multi-user collaborative scenarios.

To extract rich multi-scale features from low-dimensional features, four combinations of convolution kernel sizes—(5, 9, 5), (7, 9, 5), (7, 9, 7), and (9, 11, 9)—were selected as candidates. These combinations were tested individually during the experiments, with detailed records kept of the sensing error and runtime under different PU power levels. The results are shown in Table 1. The stride of each Conv1D layer was set to 1, the convolution kernels were initialized using the Kaiming distribution, biases were initialized to zero, and zero-padding was applied. According to the data in Table 1, the convolution kernel size combination of (7, 9, 5) demonstrated lower sensing errors across various PU transmission power levels. Consequently, this combination of kernel sizes—7, 9, and 5—was adopted. Through this approach, the local feature extraction network effectively captures critical spectrum features, thereby enhancing the sensing accuracy and overall performance of the model.

**Table 1. Perception errors for different combinations of convolutional kernel sizes.**

| Emission power (mW) | Convolutional kernel size combinations | | | |
|---|---|---|---|---|
| | (5,9,5) | (7,9,5) | (7,9,7) | (9,11,9) |
| 100 | 21.4197 | 20.3750 | 20.8789 | 20.6085 |
| 200 | 12.0170 | 11.2098 | 11.4527 | 11.5497 |
| 300 | 8.6348 | 7.9609 | 8.1137 | 8.5283 |
| 400 | 6.9266 | 6.5585 | 6.8083 | 6.3294 |
| 500 | 5.0401 | 4.9997 | 4.7079 | 5.0307 |
| 600 | 3.2703 | 3.0504 | 3.0702 | 3.3165 |
| 700 | 2.8385 | 2.6350 | 2.6513 | 2.7337 |
| 800 | 1.9356 | 1.7938 | 1.8728 | 1.8995 |
| 900 | 1.8308 | 1.6900 | 1.7595 | 1.7968 |
| 1000 | 1.4386 | 1.2857 | 1.3784 | 1.3681 |

**Table 2. Perceived errors for different BiLSTM neuron numbers.**

| Emission power (mW) | Number of BiLSTM neurons | | |
| --- | --- | --- | --- |
| | 64 | 128 | 256 |
| 100 | 21.3751 | 20.1219 | 20.2352 |
| 200 | 11.0492 | 10.7128 | 10.8458 |
| 300 | 7.6352 | 7.4202 | 7.4279 |
| 400 | 6.0255 | 5.7009 | 5.6220 |
| 500 | 4.0796 | 4.0109 | 4.0358 |
| 600 | 3.2333 | 3.1423 | 3.1982 |
| 700 | 2.7517 | 2.7063 | 2.7406 |
| 800 | 1.8453 | 1.8600 | 1.8771 |
| 900 | 1.7433 | 1.8239 | 1.8196 |
| 1000 | 1.3737 | 1.3707 | 1.3827 |

In the design of the global feature extraction network, the focus was placed on optimizing the BiLSTM (Bidirectional Long Short-Term Memory) component, while temporarily omitting the MHSA (Multi-Head Self-Attention) module. The global feature extraction network operates in tandem with the local feature extraction network to enhance the model's ability to perceive complex signal environments. The candidate neuron counts for the BiLSTM layers were set to 64, 128, and 256 to identify the optimal network configuration. The selection of these values was based on the following considerations: (1) Neuron counts of 64, 128, and 256 are commonly used in deep learning models as they strike a balance between model complexity and computational resource consumption, particularly suitable for LSTM configurations in tasks such as signal processing and spectrum analysis. (2) Gradually increasing the number of neurons (from 64 to 256) facilitates the evaluation of network performance at different scales, enabling the model to achieve an optimal balance between feature representation capability and computational cost [38]. Simulation results (Table 2) indicate that under varying PU transmission power levels, setting the BiLSTM neuron count to 128 resulted in the lowest sensing error and exhibited the best sensing performance. Consequently, the BiLSTM layer's neuron count was fixed at 128.

To enhance the model's adaptability to complex signal environments, the Multi-Head Self-Attention (MHSA) mechanism was introduced. Cognitive radio spectrum sensing tasks require the model to dynamically capture variations in signal characteristics. By leveraging a multi-head design, MHSA enables the model to capture intricate relationships between features in parallel across different positions, thereby improving the comprehensiveness of feature extraction. Compared to traditional attention mechanisms, MHSA significantly enhances the model's ability to perceive diverse signals without substantially increasing computational costs, making it well-suited for dynamic spectrum environments [39]. Simulation experiments evaluated the impact of different numbers of heads (1, 4, 8, 16) on model performance, as shown in Table 3. The results demonstrate that increasing the number of heads allows the model to better uncover complex inter-feature relationships, thereby improving sensing accuracy. Notably, when the number of heads was set to 8, the model achieved the lowest average sensing error (approximately 5.61) across most transmission power levels. It also exhibited superior performance in detection accuracy, response time, and behavioral consistency. However, increasing the number of heads also significantly raised the computational complexity. When the number of heads was increased to 16, although the model captured more feature relationships, the sensing error slightly rose to approximately 5.75, likely due to information redundancy leading to feature oversaturation. Additionally, as depicted in Fig 2, resource

**Table 3. Perceived errors for different MHSA parameters.**

| Emission power (mW) | Different MHSA parameters | | | |
|---|---|---|---|---|
| | 1 | 4 | 8 | 16 |
| 100 | 19.7176 | 19.4250 | 19.2680 | 19.5778 |
| 200 | 11.1447 | 10.9409 | 10.0616 | 10.2674 |
| 300 | 7.5732 | 7.7804 | 7.3400 | 7.5369 |
| 400 | 5.3358 | 6.3271 | 5.6905 | 5.8148 |
| 500 | 3.8785 | 4.6812 | 4.2487 | 4.3537 |
| 600 | 2.8771 | 2.9128 | 2.7808 | 2.8407 |
| 700 | 2.5751 | 2.3367 | 2.5389 | 2.6170 |
| 800 | 1.8359 | 1.7333 | 1.6326 | 1.6253 |
| 900 | 1.5444 | 1.6213 | 1.4993 | 1.5110 |
| 1000 | 1.3680 | 1.2643 | 1.1261 | 1.2636 |

consumption escalated substantially, with average inference time rising to 155.83 ms and memory usage reaching 1054.93 MB. To balance performance improvements and computational complexity, the number of MHSA heads was ultimately set to 8. This configuration ensures low sensing error while keeping computational costs within a reasonable range, meeting the real-time processing requirements of cognitive radio systems.

In terms of network architecture and training hyperparameters, the local feature extraction network utilizes a three-layer Conv1D structure. These three convolutional layers progressively extract spectral features from low to high levels, facilitating the model's ability to learn complex signal feature structures without significantly increasing computational overhead. During model training, input data first passes through the local feature extraction network (1D CNN) to extract multidimensional local features. These features are then forwarded to the global feature extraction network (BiLSTM) to capture temporal dependencies. The local and global features are concatenated in the concatenation layer to form a joint feature vector,

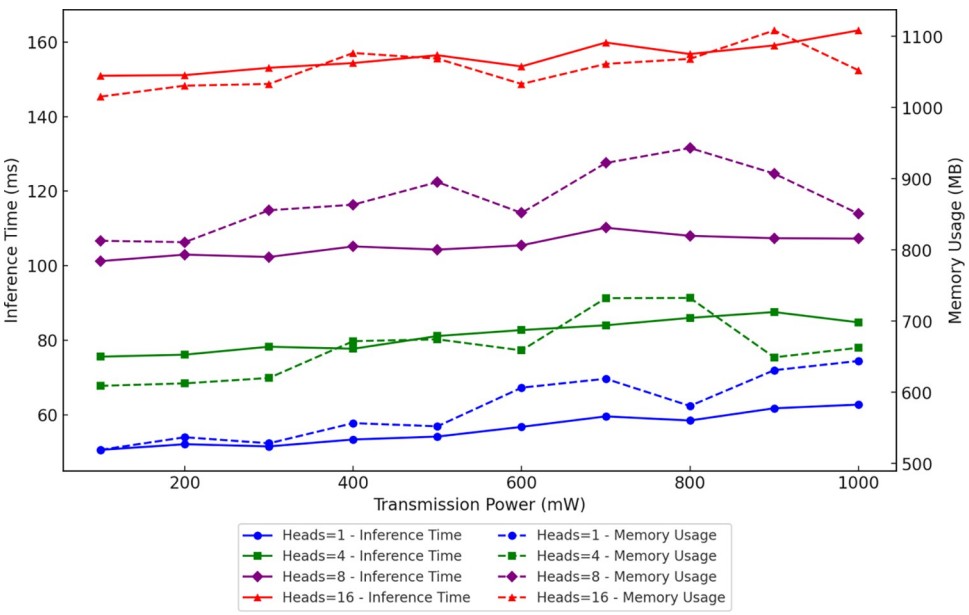

**Fig 2. Impact of MHSA heads on inference time and memory usage with distinct colors and markers.**

which is subsequently fed into the classification network for categorization. The loss function calculates the error based on cross-entropy, and parameters of both the local and global networks are optimized simultaneously via backpropagation. This joint optimization strategy ensures that the two networks complement each other in feature learning, thereby improving overall performance. Each convolutional layer employs 128 kernels, with sizes of 7, 9, and 5, respectively. This configuration balances the extraction of spectral features at different scales: smaller kernels (e.g., size 5) are effective for capturing fine-grained features, while larger kernels (e.g., size 9) are better suited for extracting broad-spectrum features. This combination enhances the model's ability to perceive multi-scale spectral features [40]. The stride is set to 1, weights are initialized using the Kaiming method, and biases are initialized to zero. Zero-padding is applied to the kernels. The activation function is LeakyReLU with a coefficient of 0.1, which helps alleviate the "dead ReLU" problem and stabilizes training by maintaining a non-zero gradient in the negative region. Each layer is followed by batch normalization (BN) and Dropout (with a rate of 0.3) to accelerate convergence and prevent overfitting [41]. The global feature extraction network's BiLSTM component contains 128 neurons, further enhancing the model's capacity to learn temporal features. The classification network is a single-layer fully connected network with 2 neurons, employing the Softmax activation function for classification. The loss function is cross-entropy, and the optimizer is Adam, chosen for its adaptive learning rate, which is particularly suitable for non-stationary data in spectral perception tasks. The initial learning rate for Adam is set to 0.003, with exponential decay factors $\beta 1$ and $\beta 2$ set to 0.9 and 0.999, respectively, and the smoothing parameter $\varepsilon$ set to $10^{-8}$. L2 regularization is also applied. The batch size is set to 200. To prevent overfitting, early stopping is used, with a patience value of 5. The specific algorithm flow is as follows:

```
# Local Feature Extraction Network
def local_feature_extractor(input_shape):
inputs = Input(shape = input_shape)
x = Conv1D(128, 7, padding = 'same', kernel_initializer = he_normal(),
bias_initializer = 'zeros')(inputs)
x = LeakyReLU(alpha = 0.1)(x)
x = BatchNormalization()(x)
x = Dropout(0.3)(x)
x = Conv1D(128, 9, padding = 'same', kernel_initializer = he_normal(),
bias_initializer = 'zeros')(x)
x = LeakyReLU(alpha = 0.1)(x)
x = BatchNormalization()(x)
x = Dropout(0.3)(x)
x = Conv1D(128, 5, padding = 'same', kernel_initializer = he_normal(),
bias_initializer = 'zeros')(x)
x = LeakyReLU(alpha = 0.1)(x)
x = BatchNormalization()(x)
x = Dropout(0.3)(x)
return Model(inputs, x)
# Combined Model
def create_model(input_shape):
inputs = Input(shape = input_shape)
x = local_feature_extractor(input_shape)(inputs)
x = Bidirectional(LSTM(128))(x)
outputs = Dense(2, activation = 'softmax')(x)
return Model(inputs, outputs)
# Parameters
input_shape = (None, 128)
model = create_model(input_shape)
```

```
optimizer = Adam(learning_rate = 0.003, beta_1 = 0.9, beta_2 = 0.999,
epsilon = 1e-8)
model.compile(optimizer = optimizer, loss = 'categorical_crossen-
tropy', metrics = ['accuracy'])
# Training
early_stopping = EarlyStopping(patience = 5,
restore_best_weights = True)
model.fit(train_data, train_labels, validation_data = (val_data,
val_labels), epochs = 100, batch_size = 200, callbacks = [early_stop-
ping], verbose = 1)
```

The design, optimization, and testing of the entire model were conducted on a workstation equipped with an Intel® Xeon® Gold 6138 CPU, 128 GB of RAM, and an NVIDIA GeForce RTX Quadro 6000 GPU. The system operates on a 64-bit Windows 10 platform and was implemented in Python using the TensorFlow 2.0.0 and Keras 2.3.1 frameworks. The training accuracy and loss variations of the proposed PCBM model during the training process are shown in Figs 3 and 4.

As shown in Figs 3 and 4, the accuracy on both the training and validation sets gradually increases as the training progresses, eventually reaching a stable state. Similarly, the training loss and validation loss exhibit a continuous downward trend, stabilizing at relatively low levels. These trends indicate that the model converges effectively, and the optimization algorithm demonstrates good adaptability. Moreover, a comparison between the training and validation curves does not reveal significant overfitting. This suggests that the regularization measures and early stopping strategy effectively prevent overfitting, ensuring an improvement in the model's generalization capability.

## 4.2. Simulation results analysis

In order to verify the effectiveness of the techniques in the local feature extraction network and global feature extraction network designed in this chapter for algorithm enhancement,

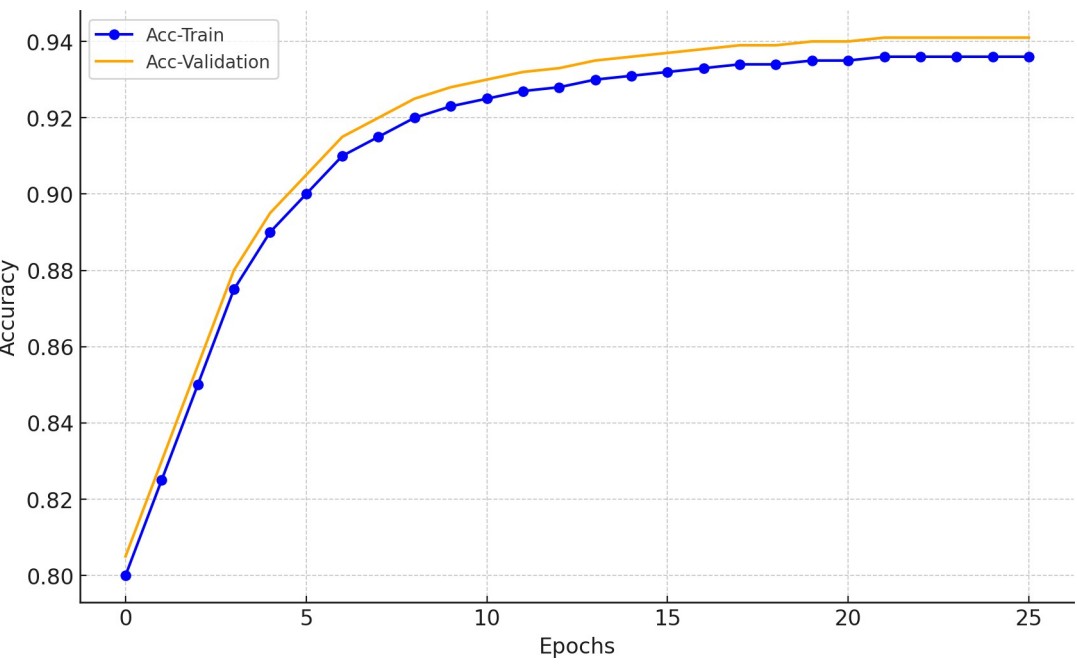

**Fig 3. Accuracy variation curve during model training.**

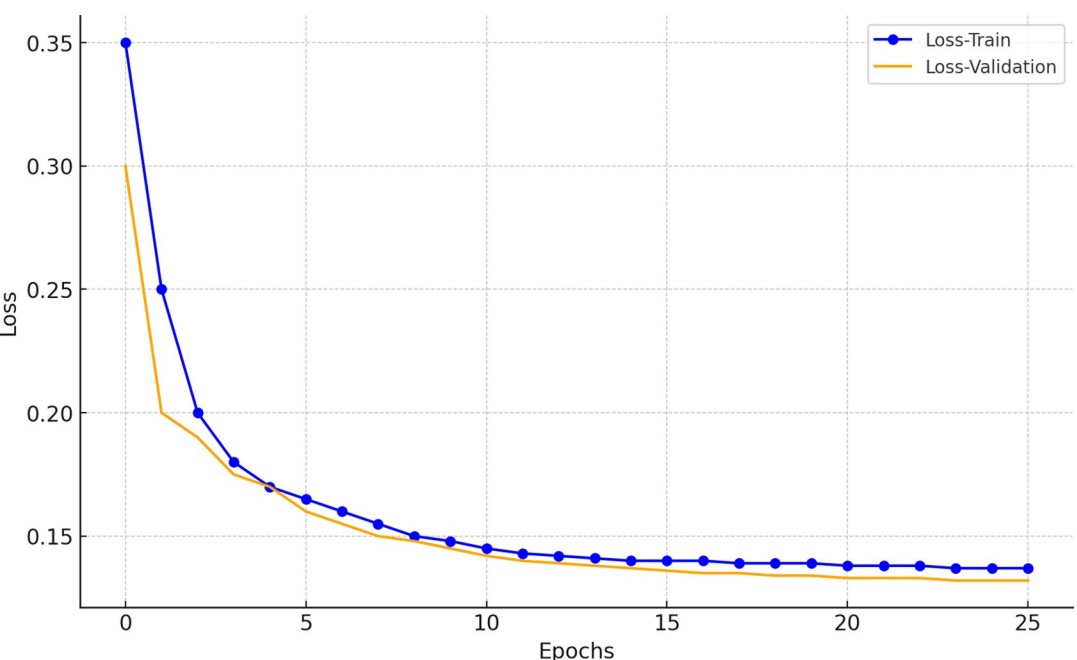

**Fig 4. Loss variation curve during model training.**

ablative simulation experiments will be carried out on the same dataset with a number of different models, and the detailed information is shown in Table 4.

In this study, the number of secondary users (SUs) in the cognitive radio system was set to 32, and the primary user's (PU) transmission power was varied from 100 mW to 1000 mW in increments of 100 mW. Under each transmission power condition, 200,000 data samples were simulated and subsequently divided into training, validation, and test sets in a 7:1:2 ratio.

As shown in Fig 5, different colors represent varying levels of spectrum utilization, with the gradient transitioning from deep purple to yellow, indicating lower to higher utilization levels, respectively. As the number of secondary users (SUs) increases from 16 to 48, the color in the heatmap gradually shifts toward yellow, signifying a significant improvement in spectrum

**Table 4. Information on different model configurations under the harmonized dataset.**

| Model name | Network type | activation function | special assembly | Number of neurons | output layer architecture |
|---|---|---|---|---|---|
| 1DCNN-ReLU | Local Feature Extraction Network | ReLU | not have | - | Single-layer fully connected network with 2 neurons |
| 1DCNN-LeakyReLU | Local Feature Extraction Network | LeakyReLU | not have | - | Single-layer fully connected network with 2 neurons |
| 2DCNN | two-dimensional convolutional network | - | Spatial correlation mining | - | - |
| CNN+LSTM | Global Feature Extraction Network | - | Single-layer LSTM | 128 | - |
| CNN+BiLSTM | Global Feature Extraction Network | - | Single-layer LSTM | 128 | - |
| CNN+BiLSTMSA | Global Feature Extraction Network | - | Bidirectional LSTM + single-headed self-attention mechanism | 128 | - |
| Our Model | PCBM Dual Network Model | Multiple activation functions | Combination of local and global feature extraction networks | manifold | manifold |

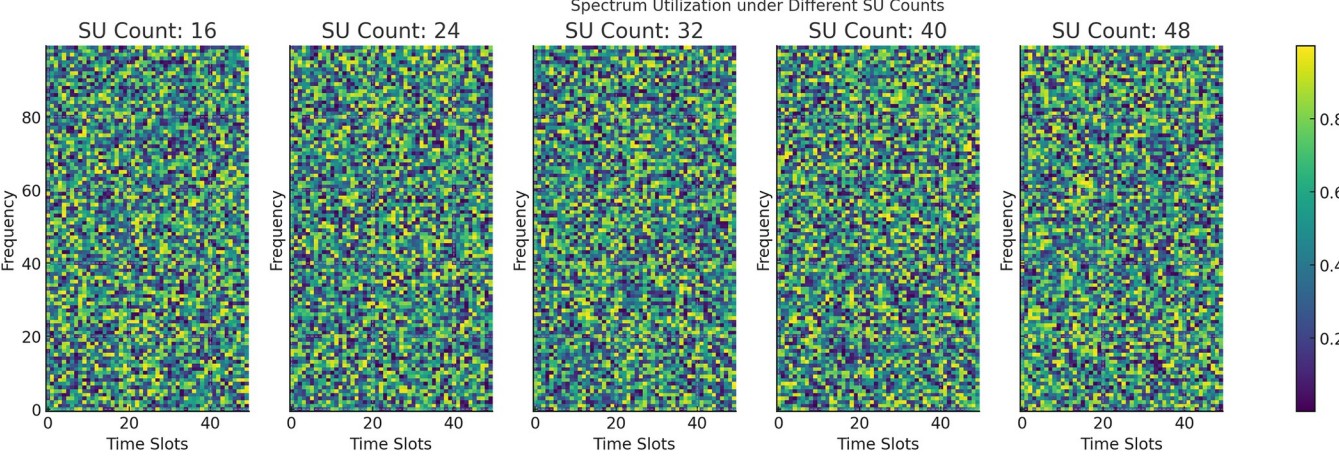

**Fig 5. Spectrum utilization under different SU counts.**

utilization. This trend suggests that a larger number of SUs can utilize available spectrum resources more effectively, thereby reducing spectrum idleness and waste, and enhancing overall spectrum efficiency. Additionally, the heatmap illustrates spectrum utilization across different time slots and frequencies, providing an intuitive view of utilization patterns. Certain time slots exhibit higher spectrum utilization, likely due to temporal variations in SU activity. For example, periods of frequent SU activity correspond to increased utilization during those intervals. Furthermore, the frequency distribution of spectrum utilization demonstrates selectivity, which could be attributed to the physical properties of specific frequency bands or the limitations of SU devices.

As shown in Fig 6, spectrum utilization exhibits a significant upward trend as transmission power increases from 100 mW to 1000 mW. This indicates that higher transmission power effectively enhances the signal coverage and spectrum sensing capabilities of secondary users (SUs), thereby improving the efficiency of spectrum resource utilization. The heatmap's color scheme transitions from deep purple (low utilization) to yellow (high utilization), clearly illustrating the increase in spectrum utilization with rising transmission power. Moreover, Fig 6 displays the distribution of spectrum utilization across different time slots and frequencies. Certain time slots exhibit higher utilization, reflecting the temporal regularity of SU activity, where frequent SU operations during these periods lead to greater spectrum resource usage. Additionally, at higher transmission power levels, spectrum utilization becomes more uniform across frequencies. This suggests that as transmission power increases, SUs achieve broader spectrum coverage, reducing resource waste caused by limited sensing capability.

## 4.3. Validity test analysis

With the experimental results of the perceptual errors of the three different models demonstrated in Table 5, we validate the effectiveness of the local feature extraction networks. The comparison results show that the perception errors of the two one-dimensional convolutional neural network (1DCNN) models are generally lower than those of the two-dimensional convolutional neural network (2DCNN) under all transmit power conditions. This finding suggests that mining spectral correlation is more effective than spatial correlation in the analysis of collaborative spectrum sensing data. This is because, when utilizing spatial correlation for collaborative spectrum sensing, the geographic locations of SUs and the relative distances between SUs must be accurately known, and the indexes of geographically neighboring SUs

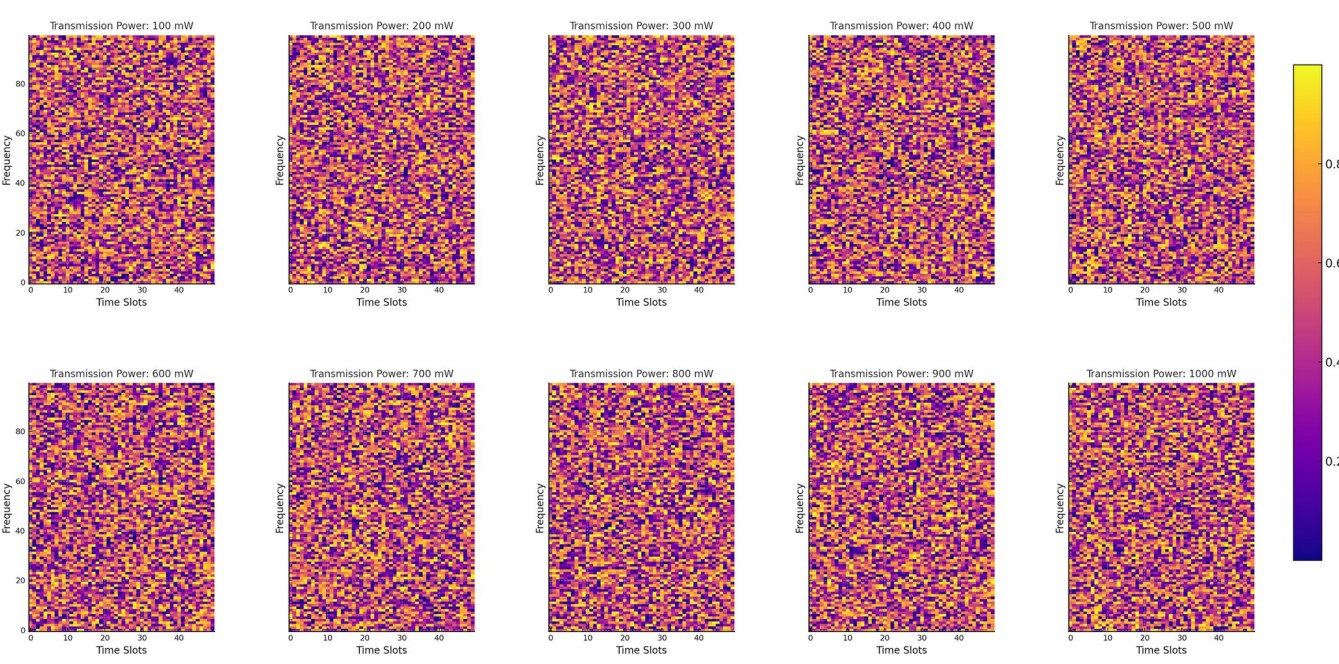

**Fig 6. Spectrum utilization under different transmission power levels.**

need to be neighboring in data indexing so that spatial correlation can work. In addition, the experiments show that the perception error of the 1DCNN model with the LeakyReLU activation function is slightly higher than that of the 1DCNN model with the ReLU activation function only at a transmit power of 500 mW. In all other test conditions, the LeakyReLU model demonstrated superior perceptual performance. This phenomenon is attributed to the fact that the LeakyReLU activation function is able to reduce the information loss and alleviate the gradient vanishing problem to a certain extent, as well as reduce the inactivity of the neurons, which effectively improves the overall performance of the model.

The experimental results presented in Table 6 and Fig 7, showcasing the sensing errors of four different models, validate the effectiveness of the global feature extraction network. Across all transmission power conditions, the PCBM model consistently demonstrates the lowest

**Table 5. Local feature extraction network ablative experimental perceptual error results.**

| Emission power (mW) | Model Type | | |
|---|---|---|---|
| | 1DCNN-ReLU | 1DCNN-LeakyReLU | 2DCNNl% |
| 100 | 22.3823 | 20.5729 | 23.9741 |
| 200 | 12.6101 | 11.3186 | 14.7376 |
| 300 | 10.0212 | 8.0383 | 10.6604 |
| 400 | 7.4437 | 6.6222 | 9.1906 |
| 500 | 4.4597 | 5.0483 | 5.8308 |
| 600 | 3.4914 | 3.0801 | 4.4268 |
| 700 | 2.7620 | 2.6606 | 3.7830 |
| 800 | 2.4147 | 1.8113 | 3.5076 |
| 900 | 2.1995 | 1.7064 | 2.5302 |
| 1000 | 1.5635 | 1.2982 | 2.1679 |

**Table 6. Global feature extraction network ablativity experiment perception error results.**

| Emission power (mW) | Model Type | | | |
|---|---|---|---|---|
| | CNN+LSTM | CNN+BiLSTM | CNN+BiLSTM-SA | Our Model |
| 100 | 20.6782 | 20.3172 | 19.5298 | 19.0845 |
| 200 | 11.2741 | 10.8168 | 11.0386 | 9.9658 |
| 300 | 8.3828 | 7.4923 | 7.5011 | 7.2701 |
| 400 | 6.5905 | 5.7563 | 5.2850 | 5.6363 |
| 500 | 4.9234 | 4.0499 | 3.8416 | 4.2083 |
| 600 | 3.4241 | 3.1728 | 2.8497 | 2.7543 |
| 700 | 3.0420 | 2.7326 | 2.5506 | 2.5147 |
| 800 | 1.9795 | 1.8780 | 1.8184 | 1.6171 |
| 900 | 1.7654 | 1.8416 | 1.5297 | 1.4850 |
| 1000 | 1.4269 | 1.3840 | 1.3550 | 1.1154 |

sensing error, particularly excelling when the PU transmission power is below 700 mW. For instance, at 100 mW, the error gap between PCBM and other models reaches up to 12%. This indicates that the PCBM model maintains superior sensing performance even under challenging wireless communication environments. Furthermore, the PCBM model outperforms existing deep learning-based CSS algorithms, such as CNN66 and LeNet-5, across all PU transmission power levels, demonstrating significant performance improvements. Specifically, compared to CNNI, which leverages spatial correlation, the PCBM model achieves lower sensing errors, highlighting its superior capability in extracting spectral data correlations. Compared with other deep learning models designed for multivariate sequential data, the PCBM model consistently achieves the lowest sensing error, showcasing its robust performance and learning capability.

## 4.4. Comparative performance analysis

To further evaluate the perceptual performance of the proposed algorithm, this section conducts a comprehensive performance comparison with 10 related algorithms. Given the limited research on collaborative spectrum sensing based on deep learning, two representative methods in this area were selected. Additionally, eight deep learning models with strong performance in handling univariate or multivariate sequence data tasks were chosen as comparative algorithms. In the multi-model performance comparison, the same dataset partitioning, transmission power range (100 mW to 1000 mW), and secondary user (SU) counts (16, 24, 32, 40, 48) were used as benchmark testing conditions. To ensure fairness and scientific rigor in the comparison, unified baseline conditions were established, and four key performance metrics were introduced: perception error, training time, testing time, and spectrum utilization. Perception error evaluates model accuracy, training and testing times measure computational efficiency, and spectrum utilization reflects the model's resource allocation capability in collaborative spectrum sensing. Furthermore, a comparative analysis of models such as CNN and LSTM was conducted to examine their characteristics in handling multivariate sequence data and to clarify the applicability of each model. The experiments were performed under identical computational resources and dataset partitions. Details of the 10 models are presented in Table 7.

In this study, specific parameters were set for the cognitive radio system, where the transmission power of the primary user (PU) was adjusted from 100 mW to 1000 mW in increments of 100 mW. A series of simulation experiments was conducted under these conditions. At these varying power levels, a total of 2 million data samples were generated, forming the

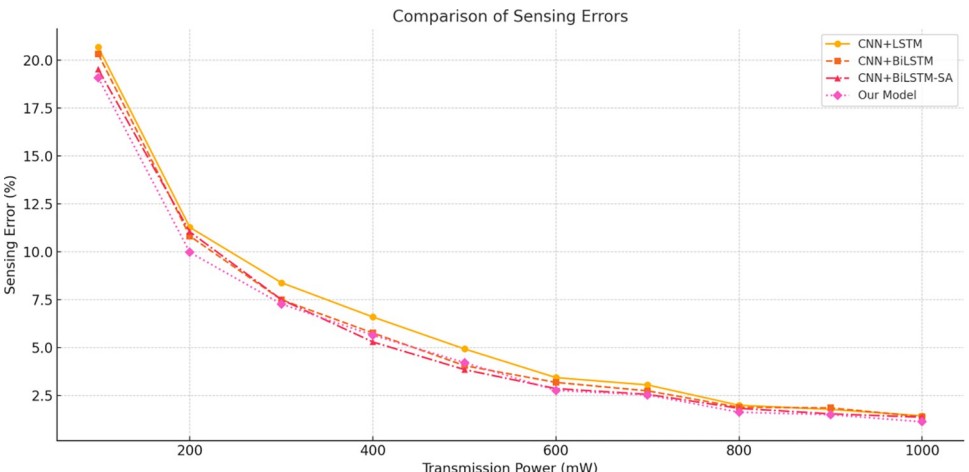

**Fig 7. Comparison of perception errors in ablation study of the global feature extraction network.**

corresponding dataset. The dataset was then split into training, validation, and test sets in a ratio of 7:1:2, respectively.

In this study, the time consumption of various deep learning models during the training and detection phases is illustrated in Fig 8, where training time is measured in seconds and detection time in milliseconds. The results indicate that convolutional neural network (CNN)-based models exhibit lower time consumption during both training and detection, with an average training time of 135.98 seconds and a detection time of 0.1056 milliseconds. This efficiency is attributed to the local connectivity and parameter-sharing characteristics of CNNs. However, the time efficiency of CNN models comes at the cost of reduced capability in handling complex data. Their performance may be limited when dealing with multidimensional

**Table 7. Detailed information of various deep learning models.**

| Model name | Data-processing capacity | feature extraction network | Optimization Features | Applicable Scenarios | Data Processing Methods |
|---|---|---|---|---|---|
| CNN | multivariate | CNN | Spatial correlation mining | Collaborative Spectrum Awareness | Spatial features are extracted using convolutional layers. |
| LeNet-5 | multivariate | LeNet-5 | - | Multi-user multi-channel CR system | A hierarchical convolutional structure is employed to capture multivariate features. |
| MLP | - | MLP | - | Benchmark model for time series classification | Fully connected layers are utilized to process multivariate sequences. |
| FCN | Single/multiple variables | FCN | - | Sequence data classification | One-dimensional convolutional layers are applied to extract local features from multivariate sequence data. |
| EncoderFCN | Single/multiple variables | FCN+SA | serial attention | Sequence data classification | A sequence attention mechanism is incorporated to enhance the capture of time-dependent features. |
| ResNet | Single/multiple variables | ResNet | residual link | Sequence data classification | Deep convolutional layers with residual connections are used to handle complex multivariate data. |
| CTN | univariate | CTN | - | Classification of time series data | Univariate Processing |
| InceptionTime | multivariate | Inception | Multi-size convolutional kernel | Serial classification | Multi-scale convolution is employed to extract features at different temporal scales. |
| LSTMFCN | Single/multiple variables | FCN+LSTM | dual network architecture | Sequence data classification | Temporal and local feature extraction are combined. |
| PCBM | - | dual network model | - | Designed for research | Joint processing of spatial and temporal features is performed. |

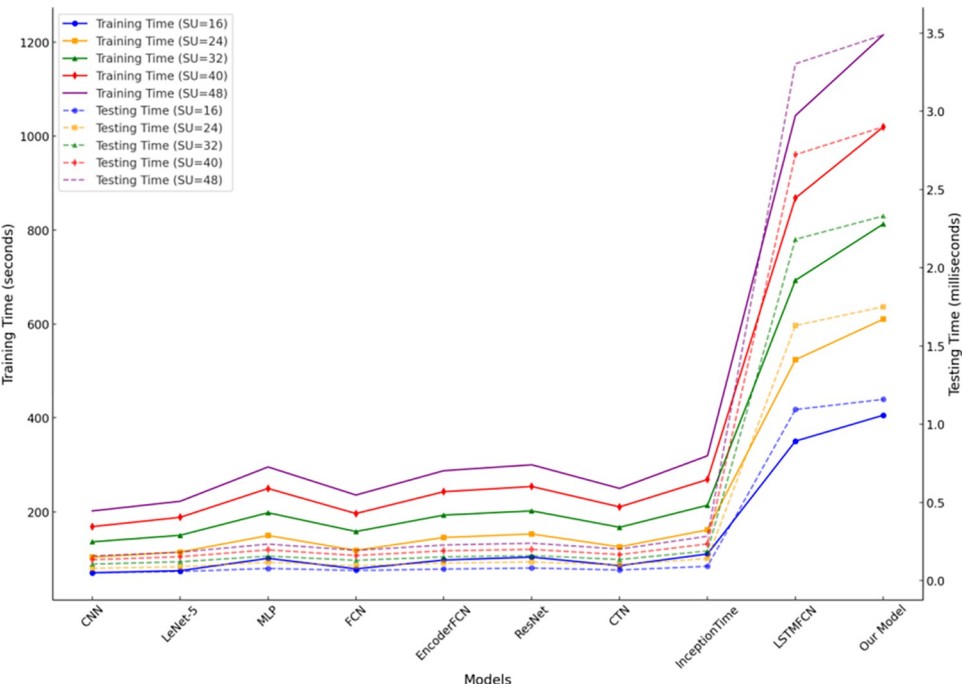

**Fig 8. Time consumption of different models.**

feature sequences or long-term dependencies. In contrast, models incorporating long short-term memory networks (LSTMs), such as LSTMFCN and the proposed model, require longer training and detection times due to the sequence-dependent nature of LSTM, which prevents parallel processing. Although the proposed model exhibits a higher detection time of 2.3262 milliseconds, this time is still in the millisecond range, which is typically acceptable in practical applications, especially in scenarios with lower real-time requirements or higher task accuracy demands.

To investigate the impact of user density on model performance, the study compared the sensing error of the proposed model with baseline models under different numbers of secondary (SUs). In the experiments, the PU's transmission power was set to 200 mW, and the number of SUs was configured at five levels: 16, 24, 32, 40, and 48. For each SU count, 200,000 data samples were simulated, resulting in a total dataset of 1,000,000 samples. The dataset was divided into training, validation, and test sets in a ratio of 70%, 10%, and 20%, respectively, and the models were evaluated under this configuration. The variation in SU numbers reflects not only the number of concurrent users in the system but also indirectly indicates the load of the cognitive radio system. Hence, studying scenarios with different SU counts helps assess the accuracy and adaptability of the models under increasing user density. The sensing errors under different SU counts are shown in Table 8 and Fig 9.

The experimental results demonstrate that as the number of secondary users (SUs) increases, the sensing error of all models decreases significantly, confirming the positive impact of higher user density on spectrum sensing performance. This trend can be primarily attributed to the increased information availability in collaborative sensing, enabling models to make decisions based on more diverse input features. Additionally, the negative influence of individual user noise is mitigated in high-density scenarios. For example, when SU = 48, the proposed PCBM model achieved the lowest sensing error of 6.2300, outperforming other complex models such as ResNet (8.2601) and InceptionTime (6.9311), and showing a more

**Table 8. Comparative experimental results with different numbers of Sus.**

| mould | SU | | | | |
|---|---|---|---|---|---|
| | 16 | 24 | 32 | 40 | 48 |
| CNN | 27.0639 | 18.5088 | 14.7376 | 11.7406 | 9.3625 |
| LeNet-5 | 28.0991 | 20.0407 | 15.0342 | 11.9215 | 8.7510 |
| MLP | 34.2809 | 24.8952 | 20.2451 | 15.8007 | 11.9953 |
| FCN | 28.3806 | 19.5295 | 12.5164 | 10.6687 | 7.9429 |
| EncoderFCN | 31.9472 | 21.3429 | 13.6161 | 11.1045 | 8.4001 |
| ResNet | 29.7637 | 19.9533 | 12.8564 | 10.8128 | 8.2601 |
| CTN | 30.6027 | 20.4270 | 13.9573 | 10.7721 | 9.0962 |
| InceptionTime | 27.2277 | 18.1932 | 11.5455 | 9.4806 | 6.9311 |
| LSTMFCN | 28.1610 | 19.4775 | 12.4816 | 10.4935 | 7.7724 |
| Our Model | 25.1659 | 16.1767 | 9.9658 | 8.4188 | 6.2300 |

pronounced advantage over traditional models like CNN (9.3625). The superiority of the PCBM model lies in its integration of the local feature extraction capabilities of convolutional neural networks (CNNs) and the global feature capturing ability of bidirectional long short-term memory networks (BiLSTMs). Furthermore, the incorporation of the multi-head self-attention mechanism (MHSA) effectively enhances the model's capacity to learn the complex relationships among variables in sequential data. Although the training time of the PCBM model exhibits nonlinear growth with the increasing number of SUs (e.g., 406.5 seconds for SU = 16 and 1219.5 seconds for SU = 48), its detection time consistently remains within the millisecond range (e.g., 3.489 milliseconds for SU = 48). This demonstrates the model's high computational efficiency and real-time applicability, making it suitable for complex scenarios with high user density. In conclusion, the proposed model not only exhibits outstanding sensing accuracy under varying user density conditions but also effectively balances computational resource consumption. This highlights its high adaptability and robustness in cognitive radio spectrum sensing tasks.

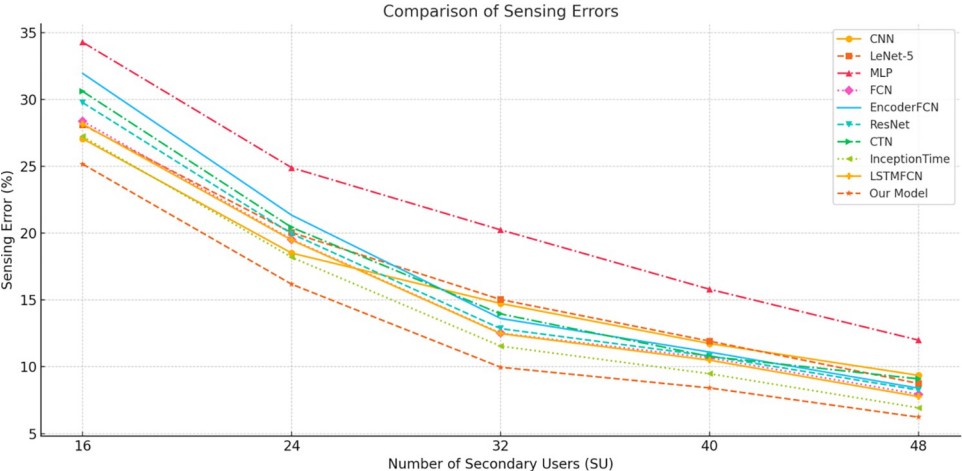

**Fig 9. Comparison of results of different methods under various SU counts.**

## 5. Discussion

This study developed a multi-user collaborative cognitive radio spectrum sensing model that integrates convolutional neural networks (CNNs) and long short-term memory networks (LSTMs), significantly improving the accuracy and efficiency of spectrum sensing. Leveraging deep learning techniques, the proposed model enables precise spectrum resource management in dynamic environments. The dual-network architecture combines the spatial sensing capabilities of the local feature extraction network (1D CNN) with the temporal dependency learning ability of the global feature extraction network (BiLSTM), demonstrating clear performance advantages in spectrum sensing tasks. Compared to traditional single-network models (e.g., models utilizing only CNNs or LSTMs), the dual-network design captures multi-scale features and long- and short-term temporal relationships in spectrum signals more comprehensively. This enhances the model's representational power and adaptability in complex signal environments. The local feature extraction network is responsible for capturing fine-grained spectral features, while the global feature extraction network further learns long-term dependencies within the time series. This design effectively improves classification accuracy. Additionally, the multi-user collaboration mechanism employed in this study significantly enhances the model's robustness and detection reliability. By integrating sensing results from multiple users, the collaboration mechanism mitigates the impact of misjudgments by individual users on overall performance, ensuring stable operation across varying noise levels and user densities. This collaborative approach aligns well with the practical requirements of cognitive radio networks, particularly in environments with high user density and elevated noise levels, where it demonstrates exceptional performance.

Compared to existing technologies, such as the CNN-based deep collaborative sensing framework developed by Lee et al. (2019), which primarily focuses on automatically learning and integrating sensing data from secondary users (SUs) within training samples, the method proposed in this study incorporates LSTMs to capture temporal dependencies, enhancing the model's adaptability to signal fluctuations. Additionally, the proposed model is designed to monitor the spectrum effectively across diverse radio environments without requiring extensive adjustments for specific scenarios. While the approach by Gao et al. (2021) improved sensing accuracy by leveraging multi-agent reinforcement learning to optimize spectrum sensing strategies, it incurred higher synchronization and communication costs. In contrast, our CNN-LSTM structure reduces the reliance on centralized processing, lowers communication overhead, and decreases energy consumption while maintaining task efficiency. Similarly, the unsupervised deep transfer learning approach employed by Li et al. (2021) improved model generalization and robustness. This finding inspires us to consider integrating transfer learning strategies into the CNN-LSTM model to adapt to dynamic radio environments [42].

Despite the model's exceptional performance across various scenarios, practical deployment poses challenges. For instance, under extremely low signal-to-noise ratios (SNRs) or non-ideal channel conditions, the model's stability and accuracy may be affected. Moreover, in resource-constrained environments—such as embedded devices or mid-range hardware platforms—computational efficiency and inference performance could become critical limitations. Future research will focus on optimizing the model's architecture, such as reducing the number of attention heads in the MHSA module or adjusting the number of neurons in the BiLSTM layers to manage computational complexity and memory requirements, thereby enhancing the model's adaptability in low-resource environments. Deployment strategies on energy-efficient platforms will also be explored to ensure effective operation under diverse hardware conditions. In larger-scale cognitive radio networks, with significantly increased SU counts or heightened spectrum environment complexity, the model's scalability is another

crucial direction for future research. Although current results indicate that the PCBM model maintains robust sensing performance as user density increases, extending it to scenarios with even higher SU densities may require further optimization of its computational efficiency and sensing capabilities. To address these challenges, future work will investigate advanced deep learning methods, such as reinforcement learning and generative adversarial networks (GANs), to enhance the model's generalization and adaptability to complex environments. Additionally, given the diverse and dynamic nature of radio spectrum environments, the application of transfer learning techniques will be explored to improve the model's adaptability and generalization across different geographic regions and spectrum conditions. These advancements will not only increase the practical deployment value of the model but also provide comprehensive technical support for future cognitive radio networks. The ultimate goal of this research is to further optimize the model and conduct application-oriented studies, enabling the model to play a broader role in future wireless communication systems, particularly in achieving efficient spectrum sharing and dynamic management, thereby driving the continuous development of spectrum sensing technologies.

## 6. Conclusion

In this paper, a cognitive radio spectrum sensing model combining convolutional neural network (CNN) and long short-term memory network (LSTM) is constructed for multi-user collaborative scenarios. The model integrates the feature extraction function of CNN and the temporal data processing advantage of LSTM, which significantly improves the accuracy and efficiency of spectrum sensing. Through experimental validation, compared with traditional methods, the model shows greater advantages in dynamic spectrum environments, especially under the conditions of dealing with multi-user interference and frequent signal changes, and is able to more accurately determine the spectrum state.

Despite good results in theoretical research and experimental validation, the model's performance in very low signal-to-noise ratio environments still faces challenges. Future research will focus on how to integrate more advanced deep learning techniques, such as augmented learning and adversarial networks, to further enhance the model's generalization ability and flexibility in adapting to the environment. Given the complexity of the radio spectrum environment, future work will also explore the performance of the model in a wider range of cognitive radio network applications, as well as the validation of its effectiveness in different geographical environments. This study not only opens up new research avenues for spectrum sensing techniques in the field of cognitive radio, but also provides strong technical support for spectrum management and dynamic allocation of resources in future communication systems. Looking ahead, it is expected that the model will play a more critical role in promoting the advancement of wireless communication technology, especially in enhancing spectrum utilization efficiency and network performance.

## Supporting information

**S1 Dataset.**
(RAR)

## Author Contributions

**Conceptualization:** Kai Wang.

**Data curation:** Kai Wang.

**Formal analysis:** Kai Wang.

**Investigation:** Yangyang Chen.

**Methodology:** Yangyang Chen.

**Project administration:** Yangyang Chen.

**Resources:** Dan Bo.

**Software:** Dan Bo.

**Supervision:** Dan Bo.

**Validation:** Shubin Wang.

**Visualization:** Shubin Wang.

**Writing – original draft:** Kai Wang, Yangyang Chen, Dan Bo, Shubin Wang.

**Writing – review & editing:** Yangyang Chen, Dan Bo, Shubin Wang.

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
