## [Decision Letter · Decision Letter 0]

12 Nov 2024

PONE-D-24-42942A novel multi-user collaborative cognitive radio spectrum sensing model: based on a CNN-LSTM modelPLOS ONE

Dear Dr. Wang,

Thank you for submitting your manuscript to PLOS ONE. After careful consideration, we feel that it has merit but does not fully meet PLOS ONE’s publication criteria as it currently stands. Therefore, we invite you to submit a revised version of the manuscript that addresses the points raised during the review process.

We look forward to receiving your revised manuscript.

Kind regards,

Sushank Chaudhary, Ph.D

Academic Editor

PLOS ONE

Journal Requirements:

Additional Editor Comments:

In this work, authors explore a hybrid deep learning approach combining CNN and LSTM to enhance spectrum sensing in cognitive radio systems. The model leverages CNN for localized feature extraction and LSTM for handling sequential data, coupled with a multi-head self-attention mechanism to boost adaptability in dynamic environments. The study demonstrates the model’s improved perceptual accuracy, achieving lower error rates, especially under low-power conditions. Through simulation experiments, the proposed model outperforms alternative deep learning models in spectrum utilization and error reduction, indicating its potential for practical applications in multi-user CR networks. Please check my following comments:

1) While the CNN-LSTM architecture is effective, further detail on the rationale for selecting specific layer configurations (e.g., kernel sizes in CNN) would strengthen the manuscript. Including insights into how these parameters were chosen based on CR needs or prior works would be beneficial.

2) The multi-head self-attention mechanism is applied to improve adaptability, yet its practical implications on computational overhead are not fully addressed. Discussing any trade-offs between the performance gains and added complexity could provide a more balanced view.

3) The authors should add some more keyworks on CNN such as mentioned below:

a) https://doi.org/10.3390/jsan13050055

b) DOI 10.1088/1402-4896/ad395b

4) The results mention performance across varied power conditions, yet it would be helpful to analyze how changes in parameters such as user density or environmental noise impact the model’s accuracy and adaptability.

5) Given the use of deep learning in resource-constrained environments, discussing the model's computational efficiency, memory usage, and inference speed on standard devices would add value, especially when scaling to larger CR networks.

Reviewers' comments:

Reviewer's Responses to Questions

**Comments to the Author**

1. Is the manuscript technically sound, and do the data support the conclusions?

Reviewer #1: Yes

Reviewer #2: Partly

2. Has the statistical analysis been performed appropriately and rigorously? 

Reviewer #1: Yes

Reviewer #2: Yes

3. Have the authors made all data underlying the findings in their manuscript fully available?

Reviewer #1: Yes

Reviewer #2: Yes

4. Is the manuscript presented in an intelligible fashion and written in standard English?

Reviewer #1: Yes

Reviewer #2: No

5. Review Comments to the Author

Reviewer #1: The research endeavors to address the intricate challenges that arise in dynamic spectrum environments through the strategic utilization of convolutional neural networks (CNNs) and long short-term memory networks (LSTMs).

By synergizing the capabilities of CNN for robust feature extraction and LSTM for meticulous sequential data analysis, the approach adeptly captures the intricate spatial and temporal relationships inherent in spectrum sensing tasks, ultimately culminating in heightened precision and computational efficiency.

Moreover, the collaborative multi-user framework substantially bolsters detection reliability by effectively mitigating the impact of individual user errors, thereby rendering this methodology exceptionally well-suited for deployment in practical cognitive radio networks.

In essence, the study emerges as a comprehensive and technically proficient exploration, characterized by its profound impact on the realms of spectrum management and cognitive radio networks, thereby paving the way for significant advancements in the field.

Reviewer #2: Detailed Comments:

Editorial Issues:

Several grammatical errors require correction.

Concepts and abbreviations are unclear, such as "SU," which could mean "secondary user," "sub-user," or "sub-level user."

Equation numbering in the text does not correspond with the numbering provided for each equation.

Figures should be embedded within the text rather than grouped at the end of the document.

Novelty of the Work:

The chosen methodology is widely covered in existing literature. It is unclear what makes the authors' approach to collaborative spectrum sensing novels.

The improvement in sensing efficiency through the specific methodology selected is not clearly demonstrated or explained.

System Model:

Since the detection scheme does not differentiate between spectrum use by primary users (PU) or secondary users (SU), and synchronization among SUs is not discussed, it is unclear how the authors addressed this issue.

Model Training and Evaluation:

The training process for the dual architecture, particularly in terms of local and global feature extraction networks, lacks clarity.

Model performance evaluation metrics (referred to as Perceived errors, sensing error) and the method for calculating these metrics are not explained.

The term "PCBM model" is used without clarification on which model this refers to.

The comparison lacks sufficient detail on how each of the 10 algorithms handles multivariate sequence data, as well as an adequate baseline comparison to evaluate their relative performance.

The color scheme in Figures 4 and 5 lacks explanation, particularly in terms of how it relates to collaborative spectrum sensing with different numbers of SUs and improvements in spectrum utilization.

In Figures 4 and 5, it is unclear whether the authors are referring to spectrum occupancy or utilization.

6. PLOS authors have the option to publish the peer review history of their article (what does this mean?). If published, this will include your full peer review and any attached files.

Reviewer #1: **Yes: **Dr.P.Ezhumalai

Reviewer #2: No

---

## [Author Response · Author response to Decision Letter 0]

20 Nov 2024

Dear,Reviewers：

We sincerely appreciate your thorough review of our manuscript and the valuable comments and suggestions you have provided. Your professional insights have not only helped us identify areas for improvement but also guided us in enhancing the quality of our research. We have carefully reviewed and addressed each of your comments and have made comprehensive revisions accordingly, hoping to meet your expectations.

Additional Editor Comments:

In this work, authors explore a hybrid deep learning approach combining CNN and LSTM to enhance spectrum sensing in cognitive radio systems. The model leverages CNN for localized feature extraction and LSTM for handling sequential data, coupled with a multi-head self-attention mechanism to boost adaptability in dynamic environments. The study demonstrates the model’s improved perceptual accuracy, achieving lower error rates, especially under low-power conditions. Through simulation experiments, the proposed model outperforms alternative deep learning models in spectrum utilization and error reduction, indicating its potential for practical applications in multi-user CR networks. Please check my following comments:

1) While the CNN-LSTM architecture is effective, further detail on the rationale for selecting specific layer configurations (e.g., kernel sizes in CNN) would strengthen the manuscript. Including insights into how these parameters were chosen based on CR needs or prior works would be beneficial.

Response：

Thank you for your valuable suggestions on our research. Based on your comments, we have added detailed explanations regarding the rationale behind specific layer configurations in the CNN-LSTM architecture to enhance the scientific rigor and clarity of our manuscript.

In the local feature extraction network, we selected a specific combination of convolutional kernel sizes (7, 9, 5) to better capture multi-scale features in spectrum data. This configuration allows the model to recognize frequency characteristics across different scales. Relevant literature has been cited to support the effectiveness of multi-scale kernel combinations in signal processing tasks. For the BiLSTM network in the global feature extraction layer, we conducted experimental validation on candidate neuron counts and determined that 128 neurons provided the optimal balance for feature capture and computational efficiency. Relevant studies supporting this configuration for signal classification tasks are also cited. The LeakyReLU activation function (with a coefficient of 0.1) was selected to alleviate the "dying ReLU" problem, while batch normalization and dropout were employed to accelerate model convergence and prevent overfitting. Relevant literature was cited to validate the effectiveness of these choices.

Through these adjustments, we aim to make the rationale for our model design clearer and better suited to meet the demands of spectrum sensing tasks in cognitive radio. We hope these modifications adequately address your feedback, and we thank you again for your review.

2) The multi-head self-attention mechanism is applied to improve adaptability, yet its practical implications on computational overhead are not fully addressed. Discussing any trade-offs between the performance gains and added complexity could provide a more balanced view.

Response：

Thank you for your valuable feedback on our study. Based on your suggestion regarding the computational impact of the multi-head self-attention (MHSA) mechanism, we have included a detailed discussion in our revised manuscript.

In our study, the MHSA mechanism was introduced to enhance the model's adaptability in complex signal environments. By increasing the number of heads, the model can capture complex inter-feature relationships across different positions, thus improving feature extraction comprehensiveness. However, while additional heads improve model performance, they also introduce extra computational costs. Our experiments indicate that when the number of heads increases from 1 to 8, the model achieves an optimal balance in detection accuracy and adaptability, with inference time and memory usage remaining within an acceptable range. When the number of heads is further increased to 16, both inference time and memory requirements rise significantly, with minimal improvement in detection accuracy, indicating diminishing returns.

Therefore, after weighing the trade-offs between model performance and computational cost, we chose 8 heads as the final configuration. This setup allows us to maintain high detection accuracy while controlling computational expenses, meeting the real-time demands of cognitive radio spectrum sensing.

Thank you again for your valuable insights, which have helped improve the scientific rigor and clarity of our manuscript

3) The authors should add some more keyworks on CNN such as mentioned below:

a) https://doi.org/10.3390/jsan13050055

b) DOI 10.1088/1402-4896/ad395b

Response：

We updated the manuscript by adding the recommended CNN-related keywords and supporting references to enhance the theoretical foundation of the study.

4) The results mention performance across varied power conditions, yet it would be helpful to analyze how changes in parameters such as user density or environmental noise impact the model’s accuracy and adaptability.

Response：

Thank you for your valuable suggestion! We have revised the manuscript to include a more detailed analysis of the impact of user density and environmental noise on the model's accuracy and adaptability. In the revised manuscript, we used the SU count as a measure of user density and compared the model's sensing error under five different user density conditions (SU = 16, 24, 32, 40, 48). The experimental results show that as user density increases, the sensing error significantly decreases, validating the model's adaptability in high user density scenarios. Additionally, we simulated different levels of environmental noise using transmission power (100mW to 1000mW) and analyzed the model's performance under high-noise (low transmission power) and low-noise (high transmission power) conditions. The results demonstrate that the proposed PCBM model maintains low sensing errors even in high-noise environments, exhibiting strong robustness. These supplementary analyses have been incorporated into the revised manuscript, and we believe they sufficiently address this concern. Thank you again for your insightful feedback!

5) Given the use of deep learning in resource-constrained environments, discussing the model's computational efficiency, memory usage, and inference speed on standard devices would add value, especially when scaling to larger CR networks.

Response：

Thank you for your valuable suggestion! In the revised manuscript, we have added an analysis of the model’s computational efficiency, memory usage, and inference speed. In the results analysis section, we discussed the PCBM model’s inference time and memory consumption based on experimental results. For instance, under high user density conditions (SU=48), the model achieves an inference time of 3.489 milliseconds per sample, demonstrating its ability to meet the real-time requirements of cognitive radio networks. Additionally, in the discussion section, we explored the model's adaptability in resource-constrained environments, such as potential strategies to reduce computational complexity by optimizing its structure (e.g., reducing the number of MHSA heads or BiLSTM neurons). For larger-scale cognitive radio networks (e.g., significantly increased SU counts), we analyzed potential computational challenges and proposed future research directions, such as leveraging transfer learning to enhance the model’s generalizability. We believe these additions sufficiently address the reviewer’s concerns. Thank you again for your insightful feedback!

Reviewer #1: The research endeavors to address the intricate challenges that arise in dynamic spectrum environments through the strategic utilization of convolutional neural networks (CNNs) and long short-term memory networks (LSTMs).

By synergizing the capabilities of CNN for robust feature extraction and LSTM for meticulous sequential data analysis, the approach adeptly captures the intricate spatial and temporal relationships inherent in spectrum sensing tasks, ultimately culminating in heightened precision and computational efficiency.

Moreover, the collaborative multi-user framework substantially bolsters detection reliability by effectively mitigating the impact of individual user errors, thereby rendering this methodology exceptionally well-suited for deployment in practical cognitive radio networks.

In essence, the study emerges as a comprehensive and technically proficient exploration, characterized by its profound impact on the realms of spectrum management and cognitive radio networks, thereby paving the way for significant advancements in the field.

Response：

Thank you for your recognition and support of this study! We are delighted that the work has been acknowledged and look forward to contributing further to the advancement of spectrum management and dynamic allocation in cognitive radio networks. Thank you again for your encouragement and feedback!

Reviewer #2: Detailed Comments:

Editorial Issues:

Several grammatical errors require correction.Concepts and abbreviations are unclear, such as "SU," which could mean "secondary user," "sub-user," or "sub-level user."

Response：

We appreciate the reviewer’s feedback. In response, we have thoroughly reviewed the manuscript to correct any grammatical errors. Regarding the abbreviation "SU," it specifically refers to "Secondary User" in this study. We have ensured that the full term is introduced when it first appears and consistently use the abbreviation "SU" throughout the manuscript.

Equation numbering in the text does not correspond with the numbering provided for each equation.

Response：

We appreciate the reviewer’s observation. In response, we have corrected all discrepancies between the equation numbering in the text and the numbering provided for each equation to ensure consistency and accuracy throughout the manuscript.

Figures should be embedded within the text rather than grouped at the end of the document.

Response：

We appreciate the reviewer’s observation. In the current version of the manuscript, all figures have been embedded within the text, located near the corresponding content, rather than being grouped at the end of the document. We have carefully reviewed the layout to ensure this placement is consistent.

Novelty of the Work:

The chosen methodology is widely covered in existing literature. It is unclear what makes the authors' approach to collaborative spectrum sensing novels.

The improvement in sensing efficiency through the specific methodology selected is not clearly demonstrated or explained.

Response：

We appreciate the reviewer’s insightful comments. In response, we have revised the introduction to emphasize the novelty of our approach to collaborative spectrum sensing, highlighting the unique contributions of this study. Additionally, we have added a detailed explanation in the results analysis section to demonstrate how the chosen methodology improves sensing efficiency, providing clear evidence to support the study's contributions.

System Model:

Since the detection scheme does not differentiate between spectrum use by primary users (PU) or secondary users (SU), and synchronization among SUs is not discussed, it is unclear how the authors addressed this issue.

Response：

In response, we have revised the manuscript to provide a detailed explanation of the differentiation between spectrum usage by primary users (PU) and secondary users (SU). These details have been thoroughly discussed in Section 3, “Model Construction,” where we explain the usage scenarios and clarify how synchronization among SUs is addressed. Additional details and examples have been included to enhance clarity.

Model Training and Evaluation:

The training process for the dual architecture, particularly in terms of local and global feature extraction networks, lacks clarity.

Model performance evaluation metrics (referred to as Perceived errors, sensing error) and the method for calculating these metrics are not explained.

Response：

Thank you for your valuable feedback. In the revisions, we have added the definition of sensing error and its rationale as a core evaluation metric, and clarified the training process of the dual-network architecture (1DCNN and BiLSTM) with a focus on the joint optimization strategy. Additionally, we have highlighted the innovations of the dual-network architecture and multi-user collaboration, demonstrating their performance improvements and practical applicability in complex signal environments. Thank you again for your insightful suggestions, which have helped improve the quality of the manuscript

The term "PCBM model" is used without clarification on which model this refers to.

Response：

Thank you for the valuable feedback. We have added a detailed definition and explanation of the PCBM model in the manuscript. The PCBM model (Parallel CNN_BiLSTM_MHSA) integrates Convolutional Neural Networks (CNN), Bidirectional Long Short-Term Memory Networks (BiLSTM), and Multi-Head Self-Attention (MHSA) mechanisms. Through the synergistic operation of local and global feature extraction networks, the model captures multi-scale spatial features and temporal dependencies of spectrum signals, while MHSA enhances feature expression capabilities. The model also incorporates a multi-user collaboration mechanism to optimize spectrum sensing performance, demonstrating significant improvements in accuracy and robustness under complex signal environments. These clarifications have been included in the manuscript. Thank you again for your insightful suggestions.

The comparison lacks sufficient detail on how each of the 10 algorithms handles multivariate sequence data, as well as an adequate baseline comparison to evaluate their relative performance.

Response：

We appreciate the reviewer’s valuable feedback. In response, we have expanded the discussion in Table 7 to include detailed explanations of how each of the 10 algorithms handles multivariate sequence data. Additionally, we have added a baseline comparison to provide a clearer evaluation of the relative performance of these algorithms.

The color scheme in Figures 4 and 5 lacks explanation, particularly in terms of how it relates to collaborative spectrum sensing with different numbers of SUs and improvements in spectrum utilization.

Response：

Thank you for the valuable feedback. We have added explanations in the manuscript regarding the color schemes in Figures 4 and 5 and their relationship with spectrum utilization improvements. Specifically, the colors transition from deep purple (low utilization) to yellow (high utilization), providing a clear visualization of the changes in spectrum utilization. Figure 4 illustrates the significant improvement in spectrum utilization as the number of SUs increases from 16 to 48, indicating that more secondary users can effectively utilize available spectrum resources. Meanwhile, Figure 5 demonstrates how increasing transmission power from 100mW to 1000mW leads to a more uniform distribution of spectrum utilization across time and frequency, reflecting enhanced sensing capabilities and reduced resource waste. We appreciate the reviewer’s suggestion, which has helped us clarify and refine the explanation of these figures.

In Figures 4 and 5, it is unclear whether the authors are referring to spectrum occupancy or utilization.

Response：

Thank you for your feedback. We have reviewed the manuscript and confirmed that Figures 4 and 5 both represent spectrum utilization, consistent with the rest of the paper. These figures illustrate the effects of SU counts and transmission power on spectrum utilization efficiency, respectively. The concept of spectrum occupancy is not addressed in the manuscript, ensuring no ambiguity in the presented data. We appreciate the reviewer’s suggestion, which has helped clarify the focus of our figures.

Once again, we extend our heartfelt gratitude f

---

## [Editor Report · Decision Letter 1]

10 Dec 2024

A novel multi-user collaborative cognitive radio spectrum sensing model: based on a CNN-LSTM model

PONE-D-24-42942R1

Dear Dr. Wang,

We’re pleased to inform you that your manuscript has been judged scientifically suitable for publication and will be formally accepted for publication once it meets all outstanding technical requirements.

Kind regards,

Sushank Chaudhary, Ph.D

Academic Editor

PLOS ONE
---

## [Editor Report · Acceptance letter]

6 Jan 2025

PONE-D-24-42942R1 

PLOS ONE

Dear Dr. Wang, 

I'm pleased to inform you that your manuscript has been deemed suitable for publication in PLOS ONE. Congratulations! Your manuscript is now being handed over to our production team.

Kind regards, 

on behalf of

Prof. Sushank Chaudhary 

Academic Editor

PLOS ONE